

# New deep data hiding and extraction algorithm using multi-channel with multi-level to improve data security and payload capacity

Hanan Hardan, Ali Alawneh and Nameer N. El-Emam

Philadelphia University, Amman, Jordan

## ABSTRACT

The main challenge in steganography algorithms is balancing between the size of the secret message (SM) that is embedded in the cover image (CI) and the quality of the stego-image (SI). This manuscript proposes a new steganography algorithm to hide a large amount of secret messages in cover images with a high degree of non-perception in the resulting images. The proposed algorithm applied a multi-channel deep data hiding and extraction algorithm (MCDHEA) based on a modified multi-level steganography (MLS) approach. This approach used a new modification of the least significant bits (NMLSB) to make it hard to extract a secret message from attackers. The secret message was distributed among n-blocks; each block was hidden into a sub-channel that included multi-level hiding and flows into the main channel. Different grayscale images were used through the levels of each sub-channel and using the color image in the last level of the main channel. The image size of the multi-level was expanded from one level to the next level, and at each level, lossless image compression using the Huffman coding algorithm was applied to enable the size of the data hiding at the next level. In addition, the encryption of secret messages and intermediate cover images based on the XOR encryption algorithm is applied before the hiding process. Finally, the number of bits to be replaced at each level for both sub and main channels was four bits per byte except at the last level of the main channel based on a new approach using a non-uniform number of bits replacements. This algorithm's performance was evaluated using various measures. The results show that the proposed technique is effective and better than the previous works concerning imperceptibility and robustness. Furthermore, the results show that the maximum peak signal-to-noise ratio (PSNR) of 61.2 dB for the payload of 18,750 bytes, the maximum video information fidelity (VIF) of 0.95 for the payload of 19,660 bytes, and the maximum structural similarity index measure (SSIM) of 0.999 for the payload of 294,912 bytes.

## INTRODUCTION

The rapid growth in communication and information technology with the unlimited use of the Internet has made information security the most important daily factor to protect information from attackers. There are two different approaches to information security.

Corresponding author
Ali Alawneh,
aalawneh@philadelphia.edu.jo

The first is cryptography, and the second is steganography, where both techniques are used to protect confidential data from attacks. Furthermore, these two methods provide a higher security level and confuse attackers.

Steganography is one of the essential techniques to hide secret data in different media like color or gray images, audio files, video files, and text files. The objectives of the steganography algorithm are:

- Hiding a high capacity of confidential data.
- Make hidden data invisible.

The hiding process of secret messages on gray or color images called cover images (CI) is performed. In contrast, the image hiding information is called a stego-image (SI) (*Subramanian et al., 2021*; *Rehman et al., 2018*).

Image steganography applies in the frequency or spatial domain (*Liao et al., 2018*). Both environments have weaknesses and strengths; they hide information in the spatial domain much better with lower computational, simplicity, and load capacity.

Many applications use various techniques based on spatial domain (*Ghosal, Hossain & Sarkar, 2021*; *Wang et al., 2020*; *Kim, Ryu & Jung, 2020*). These techniques are:

- Least significant bits (LSB),
- Laguerre transforms (LT),
- Exploiting modification direction (EMD),
- Pixel value difference (PVD),
- Pixel pair matching (PPM),
- Multi-base notation system (MBNS),
- Gray level modification (GLM).

Recently, many researchers developed analytical techniques to extract crucial hidden information from the stego-image. In this work, the proposed hiding technique works against visual and statistical attacks (*Kang et al., 2019*; *Yang et al., 2020*; *Jin et al., 2020*). The main challenge in steganography algorithms is balancing between the size of the secret message (SM) that is embedded in the cover image (CI) and the quality of the stego-image (SI). This manuscript proposes a new algorithm to hide a large-sized secret message in images with a high degree of non-perception of secret data hiding in images.

The rest of the manuscript includes the following: the second section presents the related work, and the motivation and objectives are shown in the third section. The fourth section presents a multi-channel deep data hiding and extraction algorithm (MCDHEA) based on the modified deep data hiding and extraction algorithm. The results discussions are shown in the fifth section, and the sixth section offers the conclusions and future scope.

## RELATED WORKS

Researchers modified several techniques based on single-level steganography (SLS) to ensure preservation from various attacks using classical or modified LSB method (*Arun &*

*Murugan, 2018*). The other techniques use multi-level steganography (MLS) to enhance security and increase the payload capacity.

*Elshare & EL-Emam (2018)* proposed a deep data hiding and extraction algorithm (DHEA) to hide extensive secret data in multi-level color images. This algorithm improves the multi-level steganography technique (MLS) and enhances the security and payload capacity. The best peak signal-to-noise ratio (PSNR), structural similarity (SSIM), and Euclidean norm ratios were obtained as 63.2 dB for the payload of 2,500 bytes, 0.9998 for the payload of 7,864 bytes, and 205 for the payload of 7,864 bytes, respectively. The advantage of the algorithm is that it is simply due to the use of one channel with uniform levels. In contrast, the drawback of the algorithm is that the payload capacity is limited due to the slight compression ratio. In addition, using one channel instead of a multi-channel makes the hiding process performed sequentially instead of in parallel.

*Hacimurtazaoglu & Tutuncu (2022)* proposed a deep data hiding algorithm using a video steganography application. This algorithm is based LSB technique with a poly-pattern block matrix (KBM) as the key. This key is a $64 \times 64$ pixel block matrix comprising 16 sub-pattern blocks with a pixel size of $16 \times 16$. This technique was applied to improve robustness, imperceptibility, and payload capacity. The best mean square error (MSE), SSIM, and PSNR values were obtained as 0.00066, 0.99999, and 80.01458 dB for the payload of 43,827 bytes and 0.00173, 0.99999, and 75.72723 dB for the payload of 111,616 bytes, respectively. The advantage of the algorithm increases the unpredictability and resistance against statistical and visual attacks. In contrast, the drawback of this algorithm is required a large size of the stego video stream, and the number of video cover images is easy to detect by attacks. Moreover, this algorithm uses a maximum payload of 184,320 bytes, which does not have large enough.

*Xue et al. (2018)* proposed a multi-dimensional steganographic method based on (MLS-ATDSS&NS) technique. This technique works under two steganography layers (audio and network steganography layers). This technique is used to improve the security of covert communication based on the new multi-layer steganographic method (MLS-ATDSS&NS). As a result, the best PSNR and NC were obtained as 96.15 dB and 1 for the payload of 43,069, respectively. The advantage of this technique is that it can achieve anti-detectability and robustness. The drawback of this technique is that using two layers in one channel is insufficient to embed high payload capacity and can not confuse the attackers.

*Zhang et al. (2020)* proposed a novel universal deep data hiding (UDH) meta-architecture to hide and extract the encoding of a secret image from a cover image. This architecture performs extensive analysis and validates that the achievement of deep steganography can be accredited to a frequency difference between the cover image and the encoded secret image. Furthermore, this technique improves the dependent deep hiding (DDH) pipeline to produce a novel universal deep hiding (UDH) meta-architecture. The best results of PSNR, APD (average pixel discrepancy), perceptual similarity (PS), and SSIM were obtained as 39.18, 1.98, 0.0001, and 0.992 for the payload of 512 bytes, respectively. The advantage of a novel universal deep hiding (UDH) meta-architecture is to disentangle the encoding of the secret image from the cover image and decode the secret

image using high hiding capacity with visual quality. However, using one channel with deep data hiding is insufficient to increase the capacity and speed of processing.

*Ahmad, El-Emam & AL-Azawi (2021)* constructed an improved deep data hiding and extraction algorithm (IDHEA) working on color images. This algorithm develops the hiding process by enhancing the security level and payload capacity by using a small size of the cover-image at the first level and gradually increasing the size of the cover-image from one level to the next according to the enlargement ratio. As a result, the best PSNR, Signal-to-Noise ratio (SNR), MSE, and Euclidean norm ratios were obtained as 65.8 dB for the payload of 2,500 bytes, 55.395, 0.0687 for the payload of 395,310 bytes, and 155 for the payload of 78,643 bytes. The advantage of the algorithm is that it is simply due to the use of one channel. In contrast, the drawback of the algorithm is that the security level is limited due to the use of one channel, and this approach makes the hiding process performed sequentially instead of in parallel.

*Dhall, Sharma & Gupta (2020)* developed a multi-level security algorithm that uses quantum encryption of the texts. This algorithm improves the encryption and compresses the texts using the Huffman algorithm at two levels of security. The best PSNR, entropy, and correlation were obtained as 94.7, 6.3, and 0.99994 for the payload of 36 bytes, respectively. The advantage of the algorithm is based on respectable security due to applying the Huffman algorithm. In contrast, the drawback of the algorithm is that using two levels is inappropriate for confidential data to be incomprehensible.

*Sayed & Wahby (2017)* proposed a data hiding algorithm to enhance MLS. This algorithm enhanced LSB to hide secret data in BMP images using two levels (LSB-L1 and LSB-L2). As a result, the best PSNR and MSE were obtained as 67.91 and 0.01 for the payload of 270 bytes, respectively. The algorithm's advantage is that two enhancements of LSB are applied in data hiding. In contrast, the drawback of the algorithm using two levels is inappropriate for secret data.

*Bhowal (2019)* presented multi-level audio steganography to describe a new hidden communication model in secret communication technology. This algorithm improved MLS using at least two embedding methods, so the second method is used the first method as a carrier. The best SNR was obtained as 92.54 dB for the payload of 44,100 byte. The advantage of this algorithm is to increase the level of security while transmitting secure data over community channels and also can be used to deliver two or more data hiding solutions instantaneously. In contrast, the drawback of the algorithm using two levels with one channel is inappropriate for secret data.

## OBJECTIVES AND MOTIVATION

The objectives of the proposed algorithm are to develop and improve several fundamental approaches for access to secure data. Among these approaches that have been developed are the least significant bit (LSB) technology, the multi-level steganography algorithm, and the image and text segmentation to hide secret data randomly.

The Internet has a means available to give specific people legal permission to see confidential information. However, unfortunately, other people can access this confidential data who do not have legal permission. Data encryption is the most traditional

strategy for preserving information, but this method has become easy for attackers. So the alternative way to encrypt data is to hide secret data and not arouse suspicion of its existence; this technique is often called data steganography.

The proposed steganography algorithm uses a multi-channel deep data hiding and extraction (MCDHEA) technique. This technique is applied to improve the previous work based on the (IDHEA) algorithm (*Ahmad, El-Emam & AL-Azawi, 2021*) by using two types of channels. The main channel is the first type, whereas the sub-channel is the second type. The hiding algorithm suggested in this work is based on a new modified least significate bit (NMLSB) algorithm with Huffman compressing and XOR encryption techniques.

The proposed hiding algorithm aims to:

- achieve better hiding performance with the high payload capacity,
- confuse visual and statistical attackers to misleading where the blocks of SM have existed in the main channel and sub-channels,
- make the number of sub-channels and the number of levels at each channel unknown to attackers,
- make the stego-image (SI) at the last level of the main channel to closely match its corresponding cover image (CI).

In the section "Result and Discussions," we discussed all the above aims and confirmed that all aims had been satisfied.

The motivation for proposing multi-channels instead of a single channel in this work is to avoid sequential processing (hiding confidential data from one level to the next sequentially). However, sequential processing needs much time if we hide a large amount of secret data. Therefore, in this article, we proposed a new approach based on multi-channels with multi-levels at each channel to reduce the time of hiding a high payload capacity that can be working on distributed systems.

## THE SUGGESTED HIDING AND EXTRACTION ALGORITHM

This article constructs a new data hiding and extraction technique to achieve high imperceptible secret data in the stego-image. The proposed technique is preserving the superiority of the image based on the new modification of least significant bits (NMLSB) discussed in the "Data hiding algorithm" section.

Furthermore, this technique is based on deep data hiding and extraction algorithm working on a multi-channel (MCDHEA). This algorithm is the extended version of the old techniques such as multi-level steganography (MLS) (*Mahdi et al., 2019*), deep data hiding and extraction algorithm (DHEA) (*Elshare & EL-Emam, 2018*), and improved deep data hiding and extraction (IDHEA) (*Ahmad, El-Emam & AL-Azawi, 2021*). The proposed algorithm (MCDHEA) has been implemented in two phases. The first phase applied a multi-channel deep data hiding algorithm (MCDHA). In contrast, the second phase applied a multi-channel deep data extraction algorithm (MCDEA).
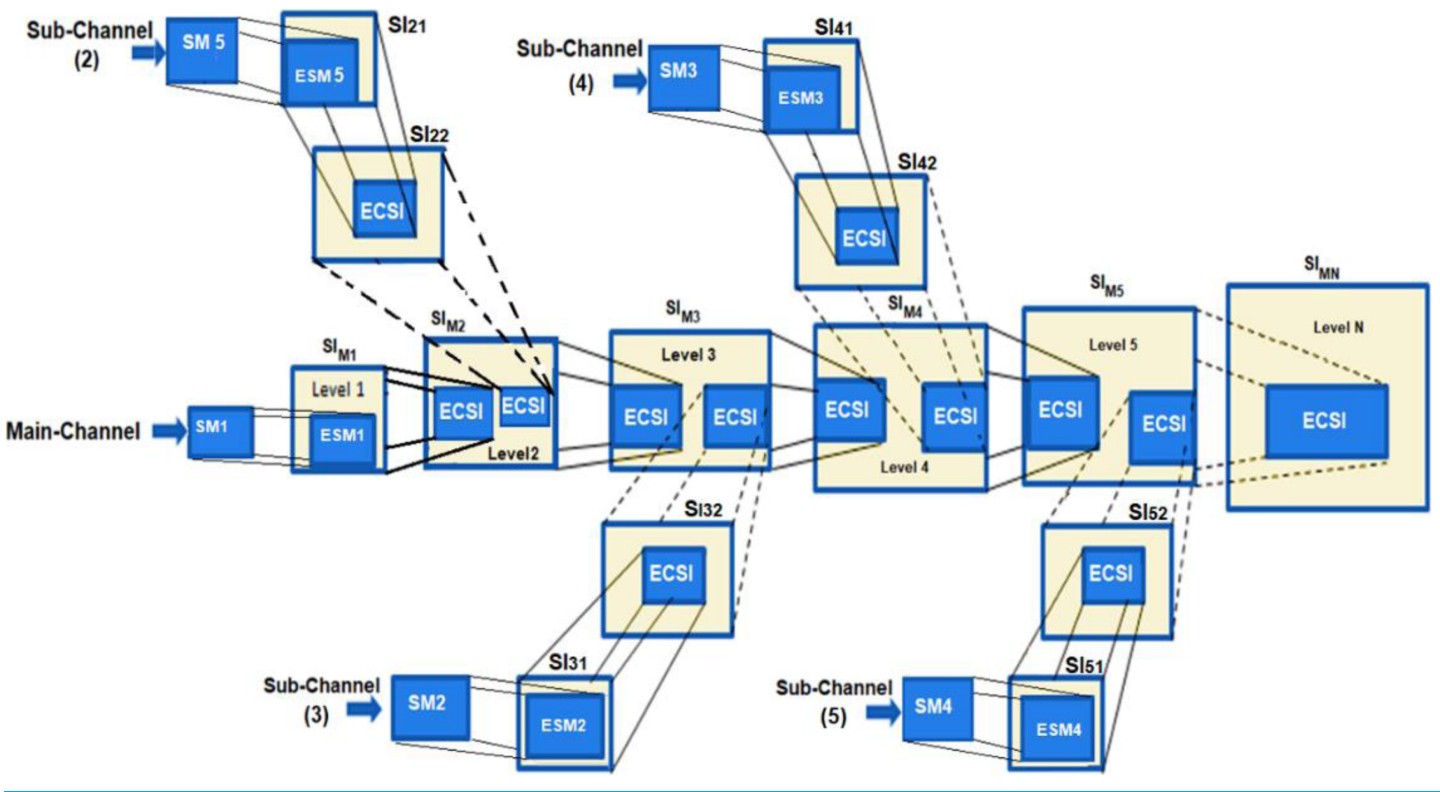

**Figure 1  Multi-channels deep hiding.**               

## Multi-channel deep data hiding algorithm (MCDHA)

The MCDHA is the modified version of the previous work (*Ahmad, El-Emam & AL-Azawi, 2021*) by dividing a secret message SM into n blocks {SM1, SM2, …, SMn}, and distributing these blocks on a multi-channel. Where SMi is the i[th] secret message hidden into a cover image of the selected channel index and level number at this channel randomly. Encryption and compression have been applied to the outputs of the hiding process to generate encryption and compression stego-image (ECSI). The process of hiding is continued through a sequence of cover images at the specific channel until the hiding process has reached the main channel. Multi-level images have been applied with progressive enlargement in image size to ensure that security and hidden data are not perceptible (see Fig. 1).

The secret message (SM) is divided into a uniform size of (n)-blocks; each block is hidden in one channel. The hiding process is performed after being encrypted by the XOR key and compressed by the Huffman coding algorithm. Then, the blocks are distributed randomly among channels using the Fibonacci algorithm (*Bommala et al., 2020*). Finally, the process of hiding is applied using a new approach (NMLSB) based on least significant bit (LSB). The proposed hiding algorithm is applied to grayscale and color images, using the image size's expansion ratio through the channel's level, see Figs. 2 and 3.

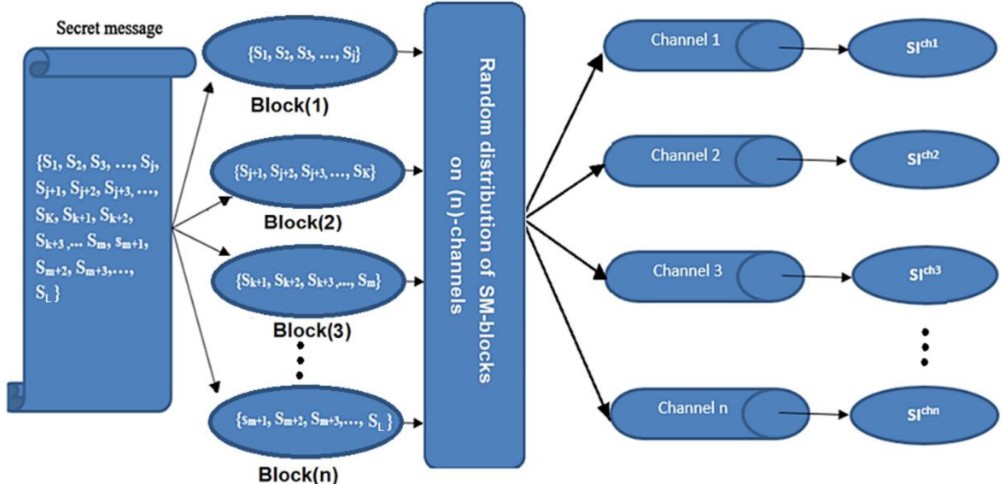

**Figure 2  Distributed a secret message randomly among channels.**

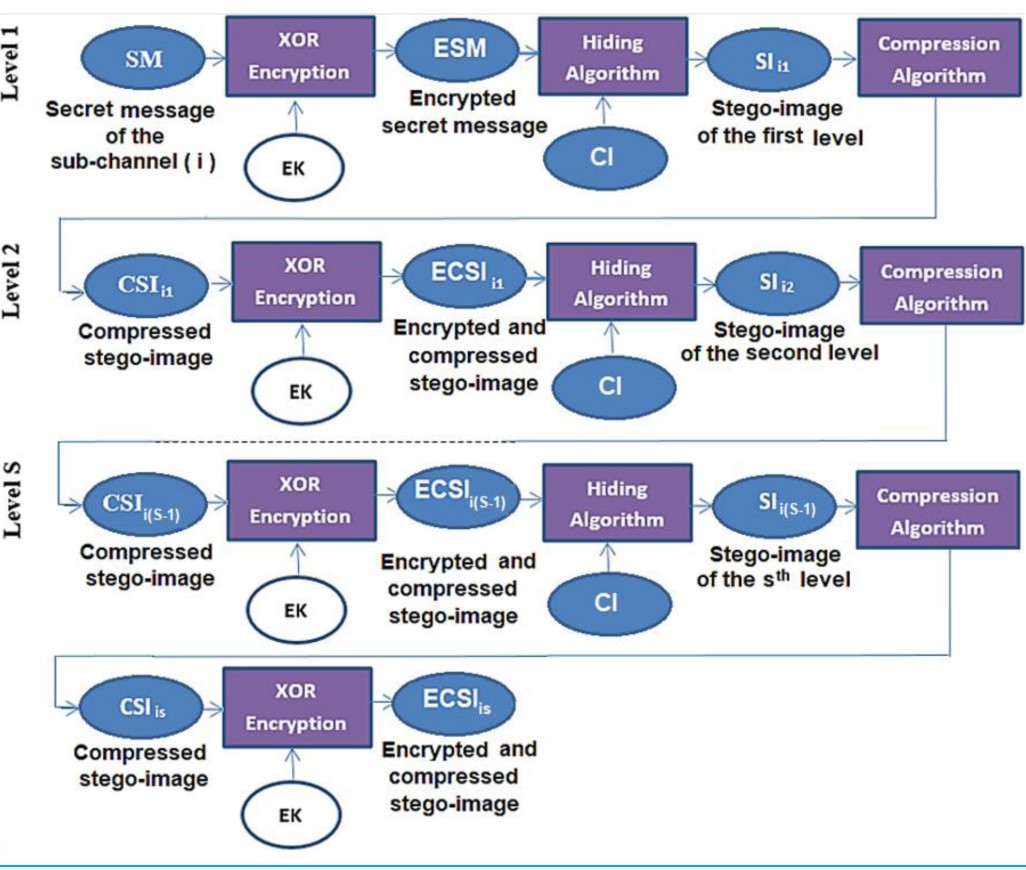

**Figure 3  Deep hiding at the i[th] sub-channel.**

The deep data hiding approach has been applied using the main channel containing N levels. For each level (i) at the main channel, the cover image at the level (i) is divided into two parts; the first part contains the hiding bits from the stego-image of

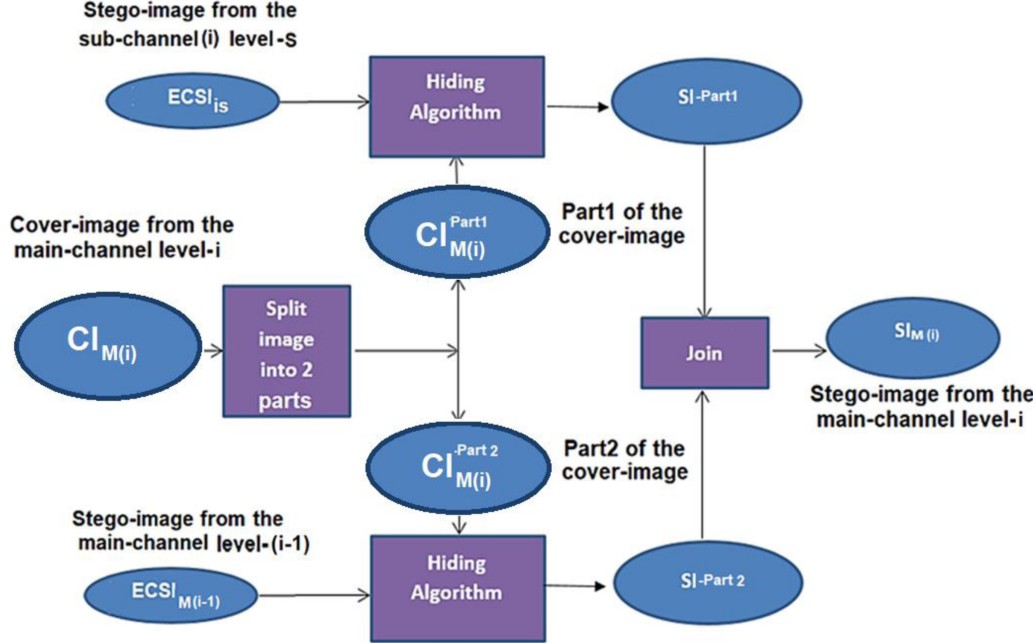

**Figure 4 Data hiding of two parts into cover-image at the main-channel using join process.**

the main channel at the level (i − 1). In contrast, the second part contains the hiding bits from a stego-image of a sub-channel. This process continues until the main channel's level (N − 1). The proposed hiding algorithm is based on the new approach using a modified LSB (NMLSB) method with encryption and compression algorithms. Finally, the two parts are joined together to generate a new stego-image created at level (i), see Fig. 4.

### The new modification of least significant bits (NMLSB)

The hiding algorithm is carried out within the chain of levels of the main and sub-channels. This process aims to produce a series of stego-images. The last level of the main channel includes a color cover image (RGB image components). This image is used to hide the stego-image from the penultimate level with the last block of the secret message to produce the last stego-image $SI_{MN}$, see Fig. 5.

Furthermore, the new modification of least significant bits (NMLSB) is based on the continuous equation to find the number of hidden bits at ith byte (i) (Nbpbi). It is calculated according to Eq. (1). Thus, it appears that the maximum value of (Nbpbi) is reached to three bits per byte if the condition $\left( \sigma^{((WSize-1)\times(WSize-1))} > 18 \text{ or } ED^{(WSize \times WSize)} > 250 \right)$ is satisfied.

$$\text{Nbpb}_i = \begin{cases} 1 \text{ if } \left( \sigma^{(WSize \times WSize)} \leq 6 \right) \\ 2 \text{ if } \left( \sigma^{(WSize \times WSize)} > 6 \text{ and } \sigma^{(WSize \times WSize)} \leq 18 \right) \\ 3 \text{ if } \left( \sigma^{((WSize-1)\times(WSize-1))} > 18 \text{ or } ED^{(WSize \times WSize)} > 250 \right) \end{cases} \quad (1)$$

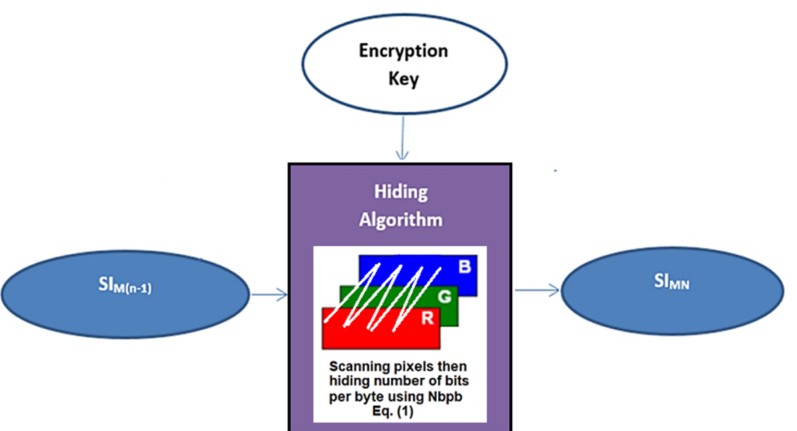

**Figure 5 Data hiding in the last level of the main-channel using RGB cover-image.**

where ($\sigma^{(\text{WSize} \times \text{WSize})}$) is the standard deviation of window size (Wsize) at the cover image when the (WSize = 3), see Eq. (2).

$$\sigma^{(\text{WSize} \times \text{WSize})} = \left( \sqrt{\frac{1}{(\text{WSize} \times \text{WSize})} \sum_{i=1}^{(\text{WSize} \times \text{WSize})} \left( CI_i - \mu_i^{(\text{WSize} \times \text{WSize})} \right)^2} \right) \tag{2}$$

and ($\mu_i^{(WSize \times WSize)}$) is the mean of window (3 × 3) at the cover image around the ith location, see Eq. (3).

$$\mu^{(\text{WSize} \times \text{WSize})}_i = \frac{1}{\text{WSize} \times \text{WSize}} \sum_{j=1}^{WSize \times WSize} CI_j \tag{3}$$

The metric ($ED^{(WSize \times WSize)}$) is the edges detection filter calculated by using Sobel edge detection filter (*Tian et al., 2021*) with window size (WSize = 3), see Eqs. (4)–(6).

$$ED_{x,CI} = \sum_{i=0}^{WSize-1} \sum_{j=0}^{WSize-1} S_j D_i * CI_{ji} \tag{4}$$

$$ED_{y,CI} = \sum_{i=0}^{WSize-1} \sum_{j=0}^{WSize-1} S_i D_j * CI_{ji} \tag{5}$$

$$ED^{(WSize \times WSize)} = \sqrt{\left( ED_{x,CI} \right)^2 + \left( ED_{y,CI} \right)^2} \tag{6}$$

where the two variables ($ED_{x,CI}$ and $ED_{y,CI}$) are calculated based on the Sobel edge detection technique based on horizontal (x-axis) and vertical (y-axis) convolution of a cover image. The Sobel operator for the x-axis is combined with optimal smoothing along the y-axis ($S_j$) with optimal differencing along the x-axis ($D_i$), see Eq. (4), while the Sobel operator for the y-axis is combined with optimal smoothing along the x-axis ($S_i$) with optimal differencing along the y-axis ($D_j$), see Eqs. (7)–(10)

$$S_i = \frac{(\text{WSize} - 1)!}{((\text{WSize} - 1) - i)!}, \ S_j = \frac{(\text{WSize} - 1)!}{((\text{WSize} - 1) - j)!} \tag{7}$$

$$D_i = P_{i,WSize-2} - P_{i-1,WSize-2} \tag{8}$$

$$D_j = P_{j,WSize-2} - P_{j-1,WSize-2} \tag{9}$$

$$P_{s,r} = \begin{cases} \frac{r!}{(1-s)! * s!} & \text{if } s \geq 0 \qquad s \leq 1 \\ 0 & \text{Otherwise} \end{cases} \tag{10}$$

where ($P_{s,r}$) is Pascal's triangle that gives sets of coefficients for a smoothing operator.

### Data hiding algorithm

The proposed data hiding algorithm (MCDHA) is described in the following steps:

## Multi-channel deep data extraction algorithm (MCDEA)

The main channel contains N levels; these are used to hide and extract (n) blocks of a secret message (SM) beside a sequence of intermediate stego-images. The extraction process has been implemented back-word from level (N) to level (1) for the main channel and from level S to level 1 from each sub-channel., see Fig. 6.

The extraction process at the ith level of the main channel produces two parts; the first part contains the stego-image of the $(i-1)^{th}$ level. In contrast, the second part contains the stego-image of the sub-channel at the specific level (i), see Fig. 7.

The secret message is extracted according to the proposed NMLSB after decompression by Huffman coding algorithm and decryption by the XOR key for the main channel and each sub-channel. See Fig. 8.

The (n) blocks of secret messages extracted from the main channel and sub-channels are joined together after reordering blocks' indexes from random distribution into ascending order of blocks' indexes using the Fibonacci algorithm21 to obtain a required secret message SM, see Fig. 9.

### Data extraction algorithm

The description of the suggested extraction algorithm (MCDEA) appears in the following algorithm steps:

### Implementation of multi-channel deep data hiding algorithm (MCDHA)

Suppose that one byte from stego-image SI is used in the $4^{th}$ level of the main channel ($SI_{M,4}$). Next, we select one byte from stego-image in the sth level of the ith sub-channel ($Sii,s$). These bytes are used to perform hiding into the cover image in the $5^{th}$ level of the main channel ($CI_{M,5}$). In this algorithm, hiding a sequence of bits from each byte into the cover image ($CI_{M,5}$) is applied to generate the corresponding stego-image ($SI_{M,5}$). The proposed algorithm uses four bits to be hidden into each byte at the cover image. As a result, it generates a stego-image at the next level of the main channel ($SI_{M,5}$). Thus, the cover image used for hiding bits is partitioned into two parts; the first part holds the hiding bits from the previous stego-image ($SI_{M,4}$). In contrast, the second part holds the hiding bits from the stego-image in the sth level of the sub-channel ($SI_{i,s}$), see Fig. 10.

| Main-algorithm: MCDHA |
| --- |

/*

*Let SM be a secret message;*

*Let N be the depth of the main channel;*

*Let S be the depth of each sub-channel;*

*Let n be the number of channels;// where n channels are used; one main-channel and n-1 sub-channels;*

*Let EK be the encryption key using the XOR approach;*

*Let SI be stego-image;*

*Let CSI be a compressed stego-image;*

*Let Ch_St_image be a stego-image from any channel;*

*Let M_Ch_St_Image be stego-image from the main channel;*

*Let RGB_CI be the color cover image;*

*Let RGB_SI be Color stego-image;*

*Let CI be a cover image;*

*Let Nbpb be the number of bits per byte;*

*/

Step 1: Input N, S;

Step 2: Compute n= N-1;

Step 3: Divide the SM into (n) blocks {SM1, SM2, …, SMn}; *//number of blocks=number of channels that include the main and sub-channels.*

Step 4: Let i←0; // i is the block's index SMi.

Step 5: Set SM←SMi;

Step 6: Compute i←i+1;

Step 7: Input CI, EK;

Step 8: Call HidingInLevel (SM, EK, CI), which returns CSI; *//hiding in level 1 at the main channel.*

Step 9: Set SM← CSI;

Step 10: Set M_Ch_St_Image ← SM;

Step 11: For (level ←2 to N-1) Do // *Passing M_Ch_St_image to the next level of the main channel starting from (level 2).*

    Step 11-1: Set SM←SMi;

    Step 11-2: Compute i←i+1;

    Step 11-3: Call HidingInChannel (SM, S) which returns Ch_St_Image;

    Step 11-4: Input CI;

    Step 11-5: Input EK;

    Step 11-6: Call JoinStegoImagesInMainChannel (M_Ch_St_Image, Ch_St_Image, CI, EK) which returns a new stego-image; // *Passing Ch_St_Image to the main channel.*

    Step 11-7: Set M_Ch_St_Image= CSI;

End for.

Step 12: Input RGB_CI, EK, Nbpb; // *calculate Nbpb according to* Eq. (1).

Step 13: Call HidingInRGB-Image (M_Ch_St_Image, RGB_CI, EK, Nbpb,) which Return SI;

    // *passing M_Ch_St_Image to HidingInRGB-Image.*

End MCDHA.

**Sub-Algorithm 1: HidingInChannel( .)**

Step 1: input SM, S

Step 2: Do

      Step 2-1: Input CI

      Step 2-2: Input EK;

      Step 2-3: Call HidingInLevel (SM, EK, CI) which return CSI;

      Step 2-4: Set SM← CSI; // passing CSI to the next level of sub-channel.

      Step 2-5: Compute S← S-1;

    While (S≠0);

Step 3: Find Ch_St_Image;

End HidingInChannel.

**Sub-Algorithm 2: HidingInLevel( .)**

Step 1: input SM, EK, CI;

Step 2: Check the type of SM, whether it is a text or an image;

Step 3: Convert SM into integer value representation;

Step 4: Find total hiding pixel and total hiding data to ensure sufficient hiding space;

Step 5: Encrypting SM using EK;

Step 6: Hiding data in CI using NMLSB Algorithm and return SI;

Step 7: Compressing SI using Huffman coding algorithm, which returns CSI;

Step 8: Output CSI;

End HidingInLevel.

**Sub-Algorithm 3: JoinStegoImagesInMainChannel( .)**

Step 1: Input M_Ch_St_Image and Ch_St_Image, CI, EK;

Step 2: Divide CI into 2 parts {part1, part 2};

Step 3: Hiding M_Ch_St_Image in part1 and Ch_St_Image in part2 using NMLSB and return a new SI;

Step 4: Compressing SI using Huffman coding algorithm, which returns CSI;

Step 5: Passing a new CSI to the next level in the main channel;

End JoinStegoImagessInMainChannel.

**Sub-Algorithm 4: HidingInRGB-Image( .)**

Step 1: input M_Ch_St_Image, RGB_CI, and EK;

Step 2: Hide the M_Ch_St_Image using an (R, G, B) order along the rows moving from left to right using the NMLSB algorithm based on Nbpb;// see Eq. (1)

Step 3: Return SI;

End HidingInRGB-Image.

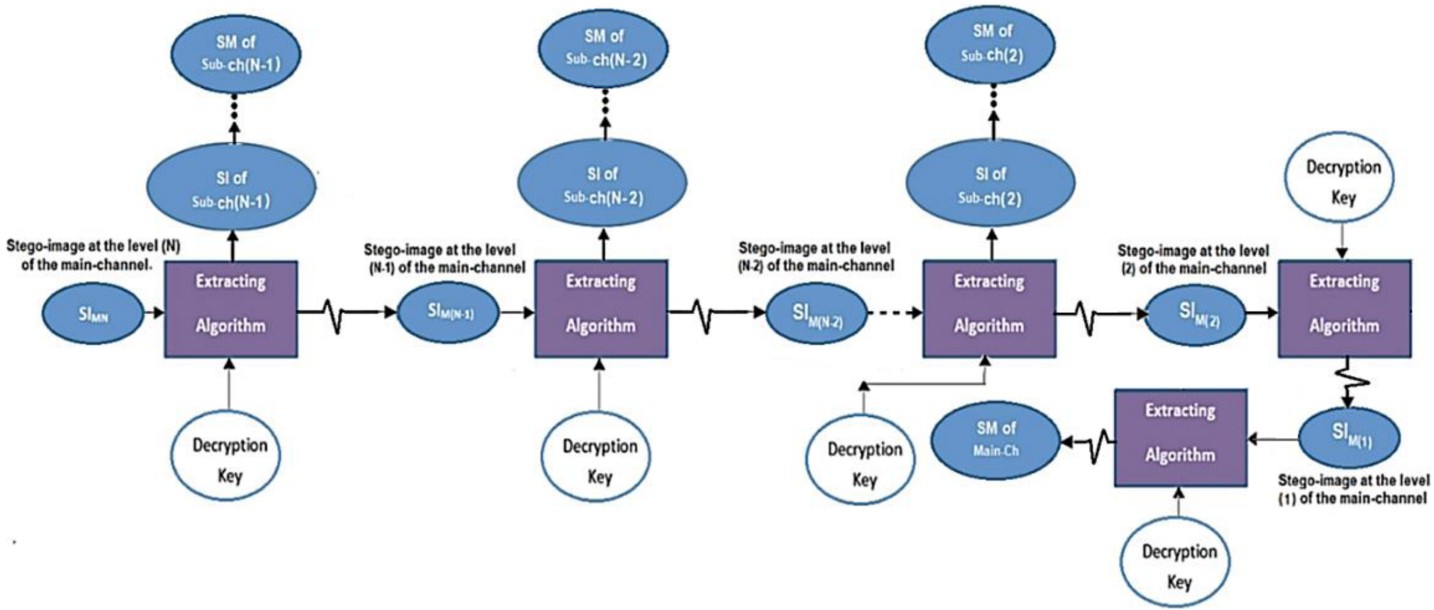

**Figure 6  Data extraction from the main and sub-channels.**

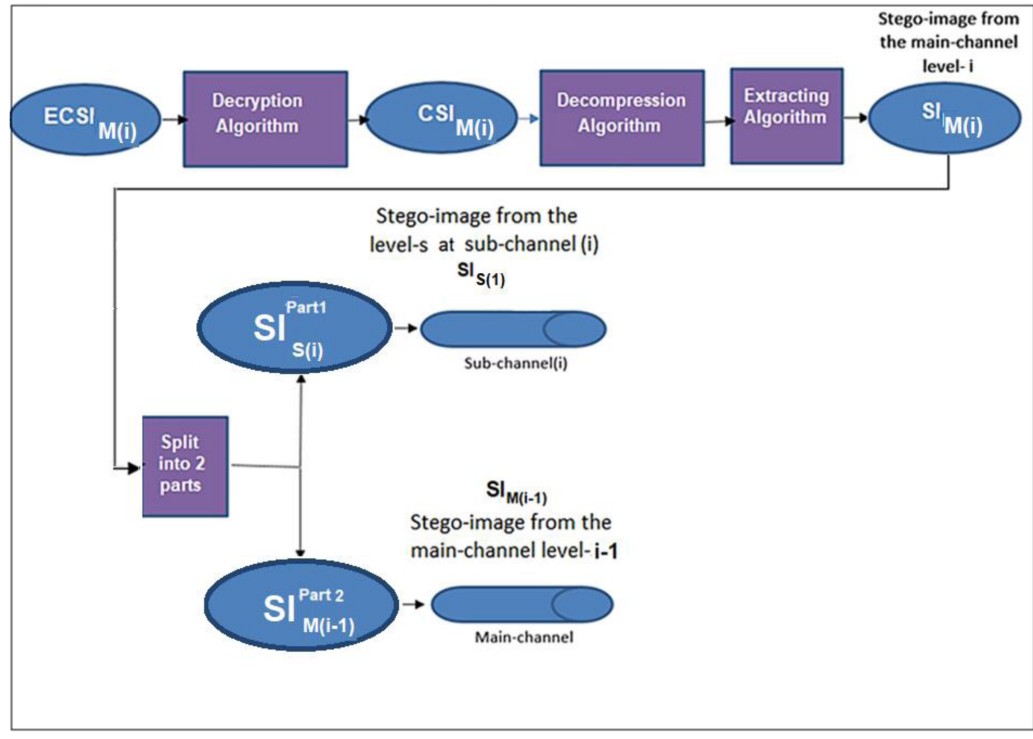

**Figure 7  Data extraction from two channels.**

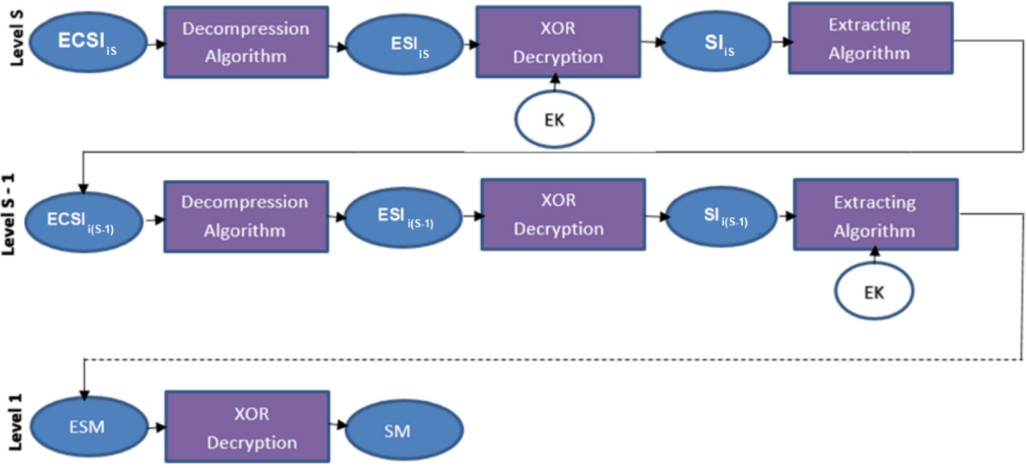

**Figure 8 Data extracting in many levels of one sub-channel.**

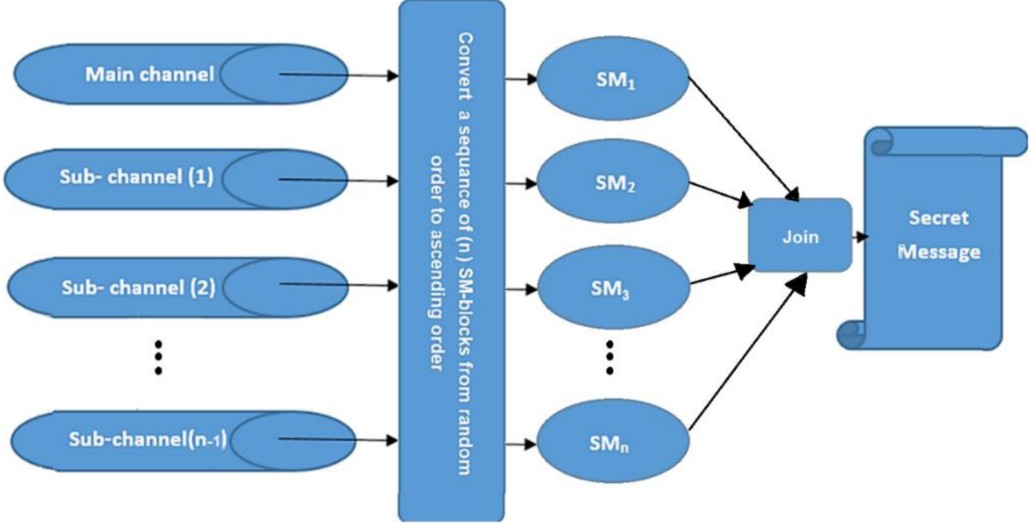

**Figure 9 Data extracting from main and sub-channels to find SM.**

## RESULT AND DISCUSSIONS

In this section, the experimental results are applied to evaluate the performance of the suggested algorithm. Many experiments have been performed using grayscale and color images with the size ($512 \times 384$ and $384 \times 512$) pixels from the UCID v2 database (*Schaefer & Stich, 2004*). In addition, standard-test-images datasets such as (Barbara, Baboon, Peppers, and Airplane) with the size ($512 \times 512$) are used to compare with the previous works. Furthermore, in this article, various measures have been discussed in the following sections to justify the security level and payload capacity.

| **Main-Algorithm: MCDEA algorithm** |
|---|

/*

    *Let SM be a secret message;*

    *Let N be the depth of the main channel;*

    *Let S be the depth of each sub-channel;*

    *Let n be the number of channels;// where n channels are used; one main-channel and n-1 sub-channels;*

    *Let DK be the decryption key using the XOR approach; // DK=EK*

    *Let SI be stego-image;*

    *Let CSI be a compressed stego-image;*

    *Let Ch_St_Image be a stego-image from any channel;*

    *Let M_Ch_St_Image be stego-image from the main channel;*

    *Let RGB_CI be the color cover image;*

    *Let RGB_SI be color stego-image;*

    *Let CI be a cover image;*

    *Let Nbpb be the number of bits per byte;// using Eq. (1)*

*/

Step 1: input CSI-RGB, DK, N, S Nbpb;

Step 2: Call ExtractFromRGB (RGB_SI, DK) which return Stego-Image; // Retrieve the CSI from RGB_SI at the last level.

Step 3: Call Extraction (Stego-Image, DK, N, S) returns the SM; // Retrieve the CSI from SI for the levels N-1 to 1.

Step 4: Extract text message from SM;

Step 5: Send a text message to the output file;

End MCDEA.

| **Sub-Algorithm 1: ExtractFromRGB ( .)** |
|---|

Step 1: input RGB_SI, and DK, Nbpb;

Step 2: Return the CSI by scanning RGB_SI using row-wise scanning from left to right using the NMLSB algorithm;

Step 3: Return CSI;

End ExtractFromRGB.

## Payload capacity *vs.* image quality

The mean square error (MSE), peak signal-to-noise ratio (PSNR), and normalized cross-correlation (NCC) are defined in Eqs. (11)–(14), and they are used to evaluate the performance of the suggested hiding algorithm.

$$PSNR = 10 * \log_{10} \frac{255 * 255}{MSE} \tag{11}$$

---

**Sub-Algorithm 2: Extraction (.)**

Step 1: Input CSI, DK, N, S

Step 2: Decompress of CSI to find SI;

Step 3: If (N=1) then

        Step 3-1: T=Decode (SI, DK);

    Else If (N=2) then

        Step 3-1: t1=Decode (SI, DK);

        Step 3-2: T=Decode (t1, DK);

        Else // Divided the SI into two parts to retrieve the secret message from each part.

        Step 3-1: sub1=SplitGray (SI);

        Step 3-2: Y2=Decode (sub1 {1, 1}, DK);

        Step 3-3: C2=Decode (sub1 {2, 1}, DK);

        Step 3-4: T2=Extraction (C2, DK, 2, S);

        Step 3-5: T1=Extraction (Y2, DK, N-1, S);

        Step 3-6: T=Strcat (t1, t2); //Merging retrieved messages from sub-channels and the main channel.

    End If.

   End If.

Step 4: Return the SM;

End Extraction.

---

**Sub-Algorithm 3: Decode ( .)**

Step 1: Input SI, DK;

Step 2: Return the encrypted message, whether it is a text or image (stego-image);

Step 3: Message decrypted using same symmetric XOR DK;

Step 4: Return SM;

End Decode.

---

$$MSE = \frac{MSE_R + MSE_G + MSE_B}{3} \tag{12}$$

where MSEi for $i \in \{R, G, B\}$ are the mean square error for the Red, Green, and Blue image colors.

$$MSE_k = \frac{1}{m * n}\sum_{i=1}^{m}\sum_{j=1}^{n}\left(CI_{ij} - SI_{ij}\right)^2; k \in \{R, G, B\} \tag{13}$$

where (i, j) is the pixel at the cover image and stego-image.

$$(NCC)_{i,j} = \frac{\sum_{i}^{m}\sum_{j}^{n}\left(\left(CI_{ij} - \mu_{CI}\right) \times \left(SI_{ij} - \mu_{SI}\right)\right)}{\sqrt{\sum_{i}^{m}\sum_{j}^{n}\left(CI_{ij} - \mu_{CI}\right)^2}\sqrt{\sum_{i}^{m}\sum_{j}^{n}\left(SI_{ij} - \mu_{SI}\right)^2}} \tag{14}$$

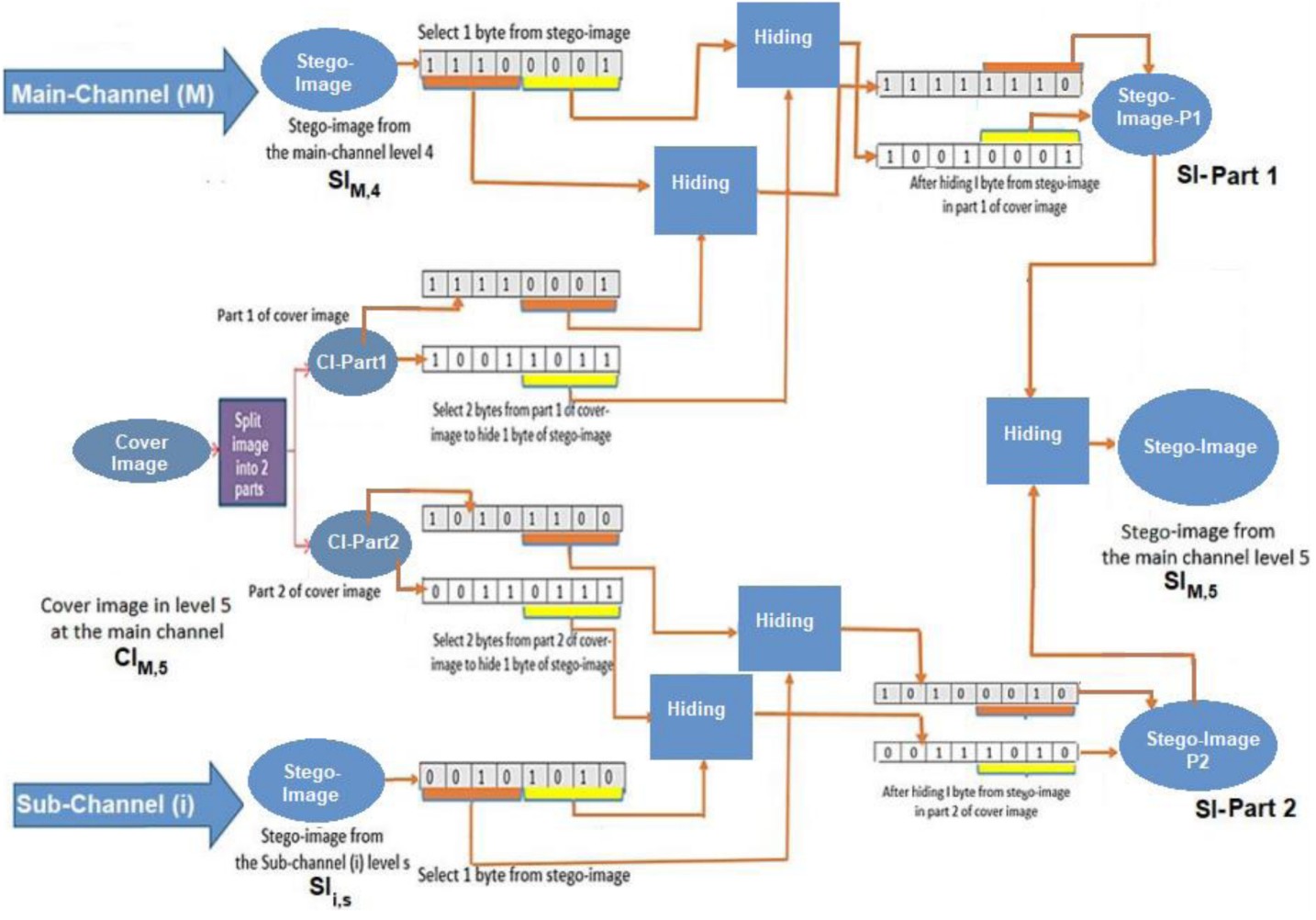

**Figure 10 Implementation of the hiding process in the main channel.**

where ($\mu_{CI}$, and $\mu_{SI}$) are the average pixels of cover and stego-images, respectively.

The proposed MCDHEA algorithm aims to make the value of (MSE) low, the value of (PSNR) high, and the value of (NCC) is closed to one. These scale values are necessary to make it difficult for the visual or statistical attacks to notice changes in the cover image. The proposed hiding algorithm is based on bit replacement using NMLSB. This algorithm is compared with previous works such as the MDLSB algorithm by *Elshare & EL-Emam (2018)*, the MLSB algorithm by *Ahmad, El-Emam & AL-Azawi (2021)*, reversible data hiding by *Ou et al. (2015)*, and the noise-like binary image blocks by *Li, Li & Yang (2013)*, see Table 1.

Moreover, the metrics MES, PSNR (dB), and NCC are calculated at the last level of the main-channel $SI_{M,N}$ of the proposed deep data hiding algorithm. According to these metrics' results, different payload capacities on three stego-images (from the standard-test-images) are applied to show the system performance. Furthermore, the proposed approach results have been compared with the previous works (*Elshare & EL-Emam, 2018*; *Ahmad,*

**Table 1 Performance of the proposed method.**

| Image size 512 × 512 | Payload capacity (bits) × 10⁵ | PSNR (dB) using MDLSB *Elshare & EL-Emam (2018)* | PSNR (dB) *Ahmad, El-Emam & AL-Azawi (2021)* | PSNR (dB) using *Ou et al. (2015)* | PSNR (dB) using *Li, Li & Yang (2013)* | Proposed MCDHEA using NMLSB | | | |
|---|---|---|---|---|---|---|---|---|---|
| | | | | | | MSE (dB) | PSNR (dB) | NCC (dB) | Diff |
| Barbara (Gray) | 2 | 63.2 | N/A | 62 | 59.8 | 0.0232 | 65.2 | 1 | 0 |
| | 5 | 59.6 | N/A | 57 | 55.8 | 0.0341 | 64.6 | 1 | 0 |
| | 7 | 58.5 | N/A | 55.1 | 54.1 | 0.0361 | 63.8 | 0.999986 | 0 |
| | 10 | 55.9 | N/A | 53 | 52.5 | 0.0404 | 62.6 | 0.999966 | 0 |
| | 12.5 | 54.8 | N/A | 51.1 | 51.1 | 0.0514 | 61.8 | 0.999934 | 0 |
| Baboon (Color) | 2 | 59.2 | 63.3 | 56.8 | 57.1 | 0.0181 | 65.6 | 1 | 0 |
| | 2.8 | 56.1 | 57.8 | 54.9 | 55.2 | 0.02504 | 64.2 | 1 | 0 |
| | 3.6 | 54.9 | 56.1 | 53.3 | 53.3 | 0.03244 | 63.0 | 1 | 0 |
| | 4.4 | 53.2 | 55.4 | 51.9 | 51.9 | 0.04012 | 62.1 | 0.999967 | 0 |
| | 5.6 | 53.4 | 53.6 | 49.8 | 49.9 | 0.05118 | 61.0 | 0.999933 | 0 |
| Peppers (Color) | 2 | 61.2 | 64.9 | 59.1 | 57.9 | 0.01222 | 67.3 | 1 | 0 |
| | 4 | 59.1 | 59.2 | 55.5 | 53.7 | 0.02122 | 64.9 | 0.999967 | 0 |
| | 6 | 55.3 | 57.5 | 53.1 | 51.8 | 0.02960 | 63.4 | 0.999967 | 0 |
| | 8 | 53.7 | 56.7 | 51.3 | 50.2 | 0.03901 | 62.2 | 0.999933 | 0 |
| | 10.5 | 53.7 | 54.9 | 49.2 | N/A | 0.05006 | 61.2 | 0.999933 | 0 |

*El-Emam & AL-Azawi, 2021*; *Ou et al., 2015*; *Li, Li & Yang, 2013*). As a result, the average PSNR for the three images using the proposed hiding algorithm is better than the results of the previous algorithms for the same images with the same data hidden capacity. Moreover, the suggested hiding algorithm that generates the "Barbara" stego-image is better than the previous works mentioned above by approximately 8.17%, N/A, 12.51%, and 14.05%, respectively. Furthermore, the Baboon stego-image generated by the proposed algorithm is better than the previous works by approximately 12.37%, 9.40%, 15.57%, and 15.35%, respectively. Finally, the "Peppers" stego-image generated by the suggested algorithm is better than the previous works by approximately 11.28%, 8.08%, 15.92%, and 16.30%, respectively. However, the present work results show the highest average of PSNR is 63.8 (dB), which appears in the "Peppers" stego-image. In contrast, the lowest average of PSNR is 63.18 (dB) in the "Baboon" stego-image. Accordingly, the present work results illustrate that when the payload sharply increases, the PSNR gradually declines. Moreover, the proposed hiding algorithm results confirm that the maximum and the minimum averages of MSE for the three stego-images are 0.037 and 0.0304 for the "Barbara" and "Peppers" stego-images, respectively. In addition, the highest average of Normalized Cross-Correlation NCC results of the three stego-images is 0.99998 for the "Baboon" stego-image.

Accordingly, the suggested hiding algorithm achieves the best performance for all payload capacities.

In addition, the difference (Diff) between the secret message before hiding (SM^H) and the secret message after extraction (SM^E) is checked in Table 1. The results of Diff

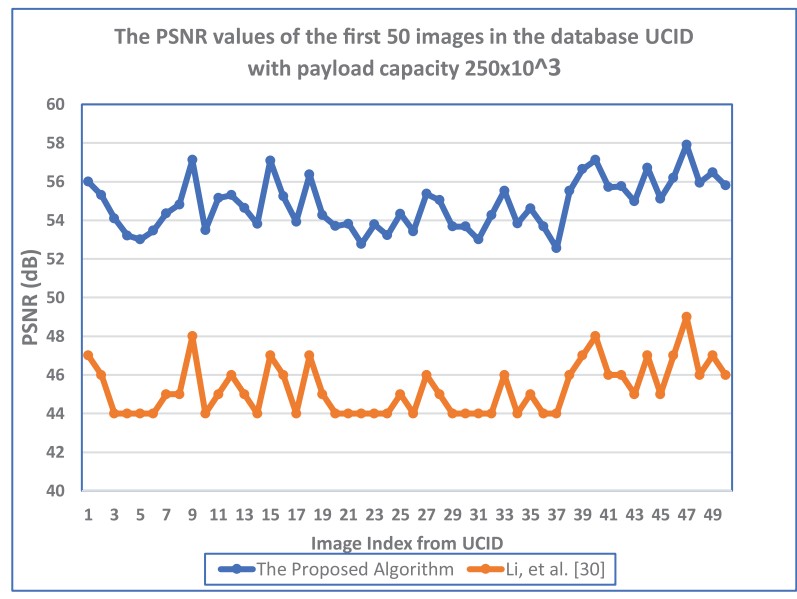

**Figure 11 The PSNR values of the first 50 images in the database UCID with payload capacity 250 × 10³ bit.** Using the orange line for (*Li, Li & Yang, 2013*) and the blue line for the 555 proposed algorithm MCDHEA.      

approved that the suggested algorithm can extract the same secret message before the hiding process by evaluating the difference according to Eq. (15).

$$\text{Diff} = \sum_{i=1}^{|SM|} \left| SM_i^H - SM_i^E \right| \tag{15}$$

where $(SM_i^H)$ is the ith character of the secret message before the hiding process. Furthermore, Fig. 11 shows the PSNR values obtained by the suggested algorithm and RDH algorithm by *Li, Li & Yang (2013)* for the first 50 images in the UCID database with the payload capacity of $250 \times 10^3$ bit. Results settle that the suggested algorithm has a more considerable PSNR value for all images. The maximum difference between the two algorithms equals 10.266 (dB). Furthermore, it appears in the image index (32). In contrast, the minimum difference between the two algorithms equals 8.547 (dB), appearing in the image index (37).

### Structural similarity index measure (SSIM)

The structural similarity index measure (SSIM) is a model of comparison metric to check the structural similarity between two images. It is calculated using Eq. (16).

$$\text{SSIM(CI, SI)} = \frac{\left(2\mu_{CI}\mu_{SI} + ((2^{24} - 1) * 0.01)^2\right)\left(2\sigma_{CI,SI} + ((2^{24} - 1) * 0.03)^2\right)}{\left(\mu_{CI}^2 + \mu_{SI}^2 + ((2^{24} - 1) * 0.01)^2\right)\left(\sigma_{CI}^2 + \sigma_{SI}^2 + ((2^{24} - 1) * 0.03)^2\right)} \tag{16}$$

where $\mu_{Ic}$ and $\mu_{Is}$ metrics represent the average of the cover and stego images, respectively, $\sigma_{IcIs}$ metric represents the covariance between the cover and the stego images, and

**Table 2 The average values of SSIM by different steganography algorithms.**

| Payload capacity | SSIM using MDLSB *Elshare & EL-Emam (2018)* | SSIM using *Li, Li & Yang (2013)* | SSIM using the proposed algorithm |
|---|---|---|---|
| 20% | 0.9997 | 0.9999 | 0.9999 |
| 30% | 0.9998 | 0.9998 | 0.9997 |
| 40% | 0.9997 | 0.9995 | 0.9996 |

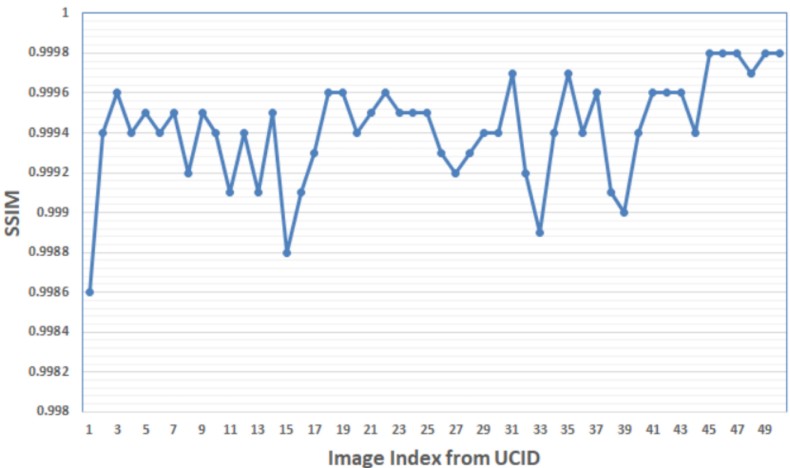

**Figure 12 The SSIM values of the first 50 images in the database UCID with payload capacity $250 \times 10^3$ bit.**

$(\sigma^2_{Ic}, \sigma^2_{Is})$ represents the variance between the cover and the stego images (*Elshare & EL-Emam, 2018*).

Structural similarity index measure (SSIM) is introduced in this study to check the image quality and attack resistance by calculating the similarity between the cove-images and corresponding steg-images. Comparisons with the previous hiding algorithms (*Elshare & EL-Emam, 2018*) and (*Li, Li & Yang, 2013*) have been made by using fifty color images selected randomly within the sizes ($512 \times 384$ and $384 \times 512$) from the UCID v2 database for three payload capacities (20%, 30%, and 40%). It found that the suggested hiding algorithm achieved excellent performance see Table 2.

Figure 12 shows the SSIM values obtained by the suggested algorithm for the first 50 images in the UCID database with a payload capacity of $250 \times 10^3$ bit. Results show that the maximum SSIM equals 0.9998, appearing in the image indexes (46–48, 49–50).

## Euclidean norm test

The Euclidean norm is defined in Eq. (17); this test demonstrates that the suggested algorithm works well against visual attacks. The distance (D) between the corresponding color {R, G, B} of both cover and stego-images is calculated.

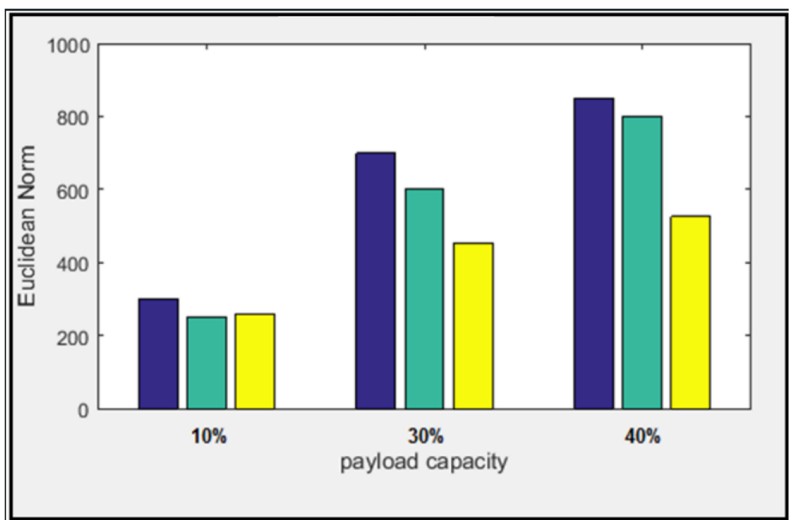

**Figure 13 Euclidean norm testing of Lena image.**

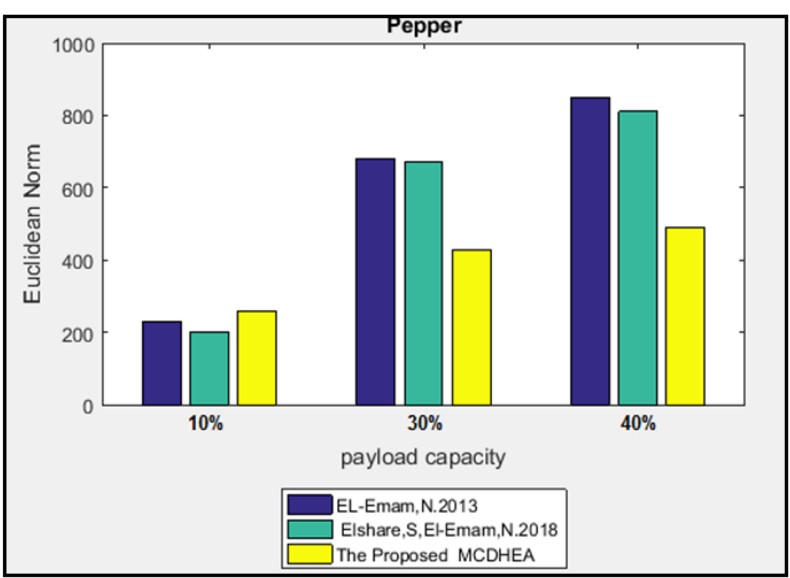

**Figure 14 Euclidean norm testing of Baboon image.**

$$D = \sqrt{(R_{CI} - R_{SI})^2 + (G_{CI} - G_{SI})^2 + (B_{CI} - B_{SI})^2} \tag{17}$$

Results of Euclidean norm (D) appeared in Figs. 13 and 14 for two-color images with image size (512 × 512) with the payload percentages are 10%, 30%, and 40%. The purpose of the suggested algorithm is to compare the results with the previous works: the MDLSB algorithm (*Elshare & EL-Emam, 2018*), and (*El-Emam & Al-Zubidy, 2013*).

The Euclidean norm distances of two images have been calculated individually; the minimum and the maximum average of the norm for three payloads are 392 and 413 at the "Peppers" and "Baboon" images, respectively. In addition, the smallest and the most

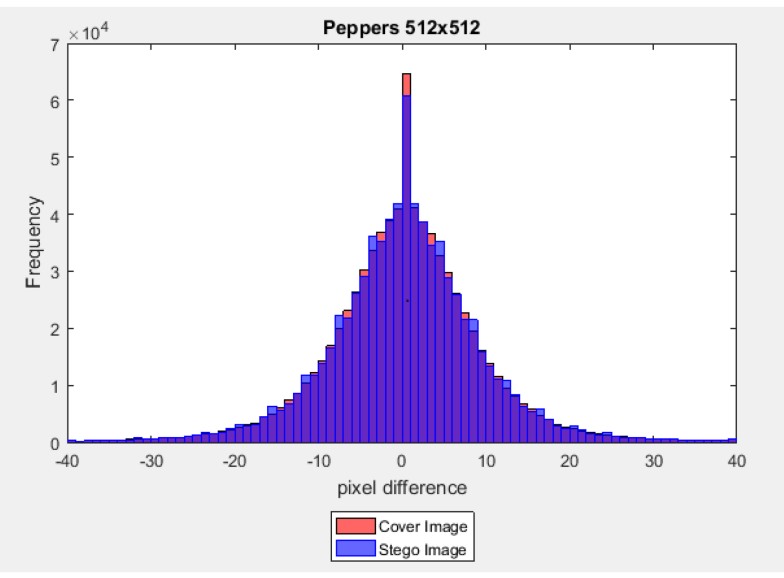

**Figure 15 Euclidean norm testing of Peppers image.**

significant difference between the average norm of the proposed algorithm and the others are 137 and 203 at the "Baboon" image for the works (*Elshare & EL-Emam, 2018*) and (*El-Emam & Al-Zubidy, 2013*) respectively.

Furthermore, the most significant difference in the Euclidean norm reached D = 360.35 at the image "Pepper," with a load ratio of 40%. It appeared between the proposed algorithm and the previous work (*El-Emam & Al-Zubidy, 2013*). At the same time, the slightest difference in the Euclidean norm reached D = 9.51 at the image "Baboon" with a load ratio of 10%. It appeared between the proposed algorithm and previous work (*Elshare & EL-Emam, 2018*).

## The difference between adjacent pixels

The difference between adjacent pixels of a cover image and its corresponding stego image was calculated by using Eqs. (18) and (19), where the scales ($D_{i,j}^{CI}$ and $D_{i,j}^{SI}$) represent the absolute difference in the adjacent horizontal pair to CI and SI (*El-Emam, 2015*).

$$D_{i,j}^{CI} = |CI_{i,j} - CI_{i,j+1}| \tag{18}$$
$$D_{i,j}^{SI} = |SI_{i,j} - SI_{i,j+1}| \tag{19}$$

where ($CI_{i,j}$ and $SI_{i,j}$) are two pixels at the location (i, j) in the cove-image and the corresponding stego-images, respectively. In this test, three-color stego-images with size (512 × 512) from the last level of the main channel with a payload capacity equal to 40% of the image size are used. The difference values ($D_{i,j}^{CI}$ and $D_{i,j}^{SI}$) belong to the interval [−255, +255], and the frequency of each of these different values was counted. Figures 15 and 16 illustrate a graph of the pixel difference value on the X-axis and the frequency on

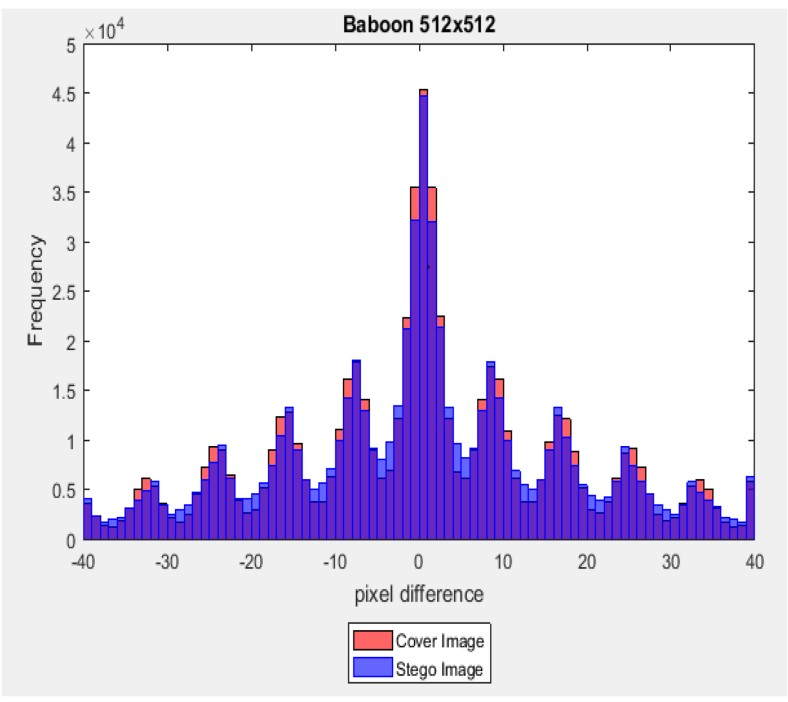

**Figure 16 Dissimilarity between adjacent pixels with payload 40%.**

the Y-axis. Results show that the difference value between the stego-image and the cover image is closed.

## Using the visual information fidelity (VIF) model

The VIF test is used to assess the loyalty of the stego-image to the corresponding cover image. This test is based on the reference field (RD), natural scene statistics NSS, and the mixture of Gaussian scale (GSM) with the image distortion (ID) and the human visual system (HVS) (*Han et al., 2013*) and (*El-Emam, 2015*). VIF test is calculated using Eq. (20), where this equation includes two mutual information measures. The first measure of the VIF test is based on the information exchanged between inputs and outputs of HVS channels without channel distortion. In contrast, the second measure of the VIF test is based on the information exchanged between the inputs and the outputs of the distorted HVS channels. The output of the HVS channels is a stego-image.

$$
\text{VIF} = \frac{\sum_{j \in \text{sub\_band}} \sum_{i \in \text{block}} \log_2 \left( \frac{\left( \sigma_{ji}^{\text{CI,SI}} \right)^2}{\left( \left( \sigma_{ji}^{\text{SI}} \right)^2 \times \left( \sigma_{ji}^{\text{CI}} \right)^2 - \left( \sigma_{ji}^{\text{CI,SI}} \right)^2 + \sigma_\mu^2 \times \left( \sigma_{ji}^{\text{CI}} \right)^2 \right)} + 1 \right)}{\sum_{j \in \text{sub\_band}} \sum_{i \in \text{block}} \log_2 \left( \frac{\left( \sigma_{ji}^{\text{CI}} \right)^2}{\sigma_\mu^2} + 1 \right)}
\tag{20}
$$

**Table 3 The average values of VIF for three stego-images.**

| Color stego-image 256 × 256 | Payload capacity % | ANN_MPSO (El-Emam, 2015) | The proposed approach |
|---|---|---|---|
| Barbara | 10% | 0.91 | 0.96 |
| Pappers | 20% | NA | 0.92 |
| Baboon | 30% | 0.86 | 0.90 |

where ($\sigma_{ji}^{CI}$ and $\sigma_{ji}^{SI}$) are the standard deviation of the cover image (CI) and stego-image (SI) in the ith block at the jth sub-band, respectively. The covariance of CI and SI in the ith block at the jth sub-band is defined in ($\sigma_{ji}^{CI,SI}$) is, see *Han et al. (2013)*.

In Table 3, the VIF measure is reported to demonstrate visual information fidelity. In this work, testing on three images from the database of the standard color images (*Imtiaz, 2019*) with the sizes (256 × 256) pixels are applied. The proposed hiding process is studied using the payload capacities and the VIF metric. The measurement of results confirmed that the suggested algorithm is better than the typical reference (*El-Emam, 2015*). The suggested algorithm is working adeptly.

## Testing chi-square (χ2) attack

The primary purpose of the suggested hiding algorithm is to hide a secret message SM in the color image without realizing the presence of hidden data. Chi-square test is used in this situation to check color uniformity in stego-images by verifying how the expected ($E_i$) and observed ($O_i$) frequencies of stego pixels are structured (*Ahmad, El-Emam & AL-Azawi, 2021*).

$$\chi_{s-1}^2 = \sum_{i=0}^{s-1} \frac{(O_i - E_i)^2}{E_i} \tag{21}$$

where (s) is the number of classes in the stego-image, (s − 1) is a degree of freedom, and ($E_i$) is the expected frequency of ($i^{th}$) pair see Eq. (22).

$$E_i = \frac{1}{2} \underset{\forall \text{ color}}{\text{fr}} \{P_{2i}, P_{2i+1}\}, \forall i = 0, \ldots, s-1 \tag{22}$$

where, $\{P_{2i}, P_{2i+1}\}$ is the $i^{th}$ pair of pixels $\{P_0, P_1, \ldots, P_{255}\}$. The frequency observed at the ($i^{th}$) color is shown in Eq. (23).

$$O_i = \text{fr}(C_i) \, \forall i = 0, \ldots, s-1 \tag{23}$$

The probability ($Pr_{\chi^2,s-1}$) based on Chi-square value ($\chi^2$) with (s − 1) degree of freedom is calculated using Eq. (24).

$$Pr_{\chi^2,s-1} = \left(2^{\frac{s-1}{2}} \Gamma\left(\frac{s-1}{2}\right)\right)^{-1} \int_{\chi^2}^{\infty} (X)^{\frac{s-1}{2}-1} e^{-\frac{X}{2}} \, dX \tag{24}$$

where the Gamma ($\Gamma(.)$) that appears in Eq. (24) is a factorial function, see Eq. (25).

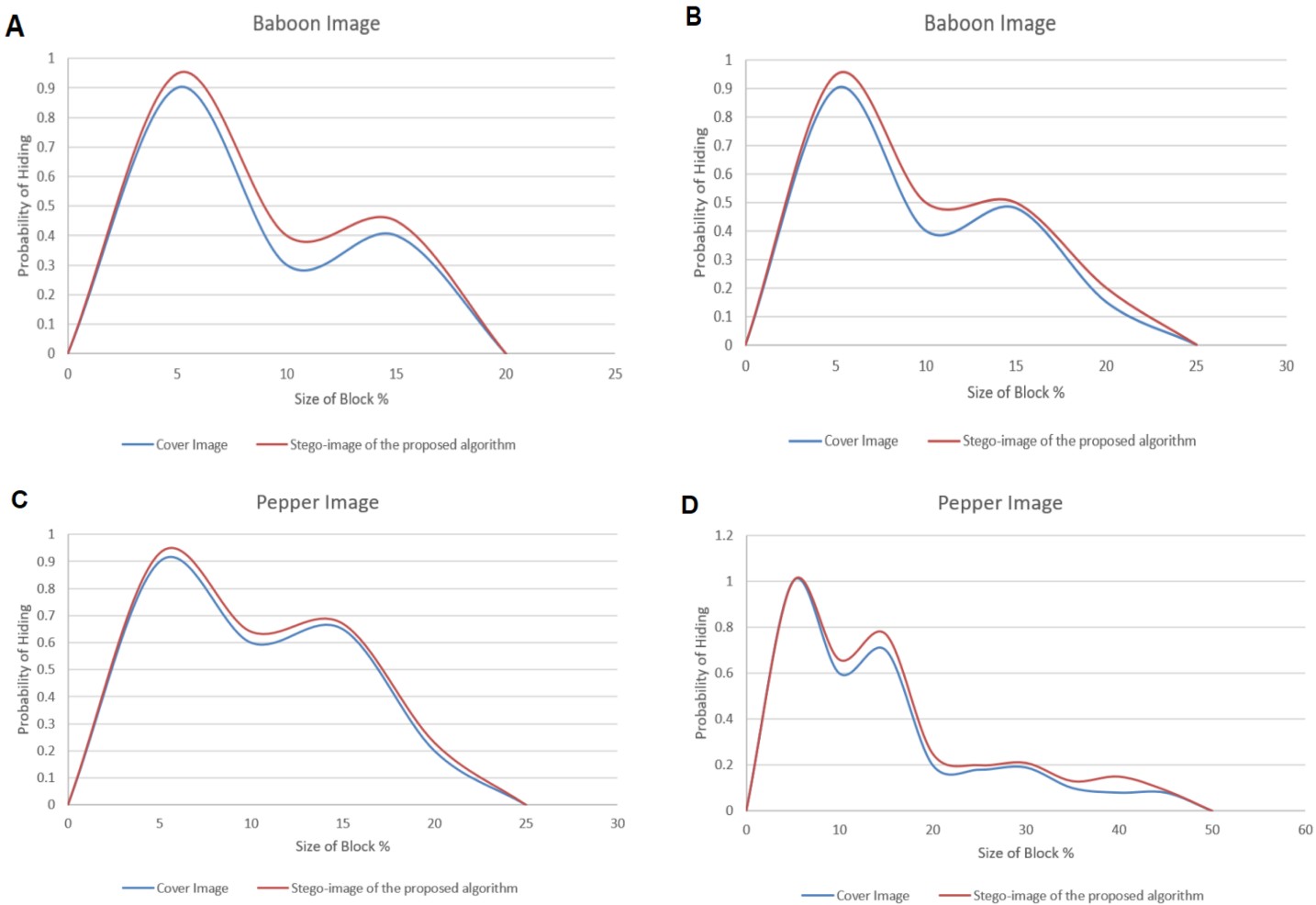

**Figure 17 The dissimilarity between adjacent pixels with a payload of 40% for Baboon and Pepper images.** (A–B) are the probability of hiding a secret message (Sm) of length 20% and 25% of the Baboon image size, respectively. (C–D) are the probability of hiding a secret message (Sm) of length 25% and 50% of the Pepper image size, respectively.

$$\Gamma(Z) = \int_0^\infty X^{Z-1}e^{-X}dX \tag{25}$$

The Chi-square attack on a stego-image with randomly distributed secret messages was studied. The proposed hiding algorithm verified comparisons between the two cover images (Baboon and Peppers) and their corresponding stego-images. The probability (Pr) values at the beginning of the block at the color image are irregular. With the block size increasing, the Pr value eventually drops to zero.

Figures 17A–17D illustrates the probabilities' results concerning the block sizes. The maximum block size reached 20 and 25 for the Baboon image and 25 and 50 for the Peppers image when the secret message length equals 25% and 50%, respectively.

Therefore, steganalysis cannot detect SM due to the identical Pr for the stego-images and their corresponding cover images.

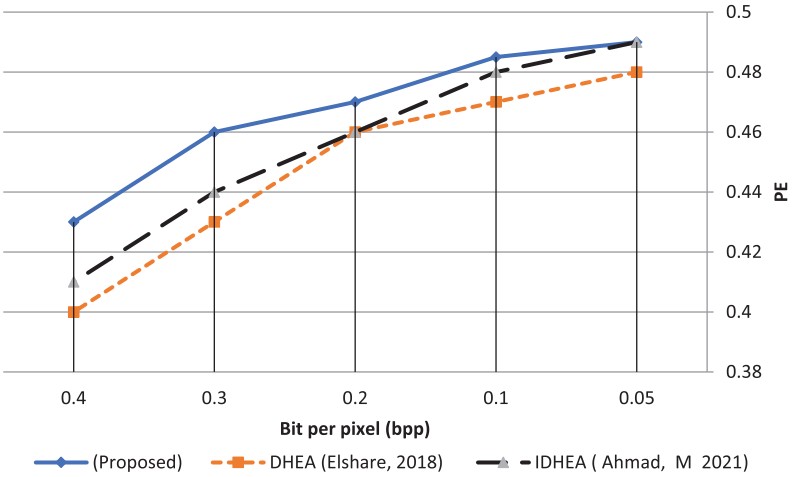

**Figure 18 Dissimilarity between adjacent pixels with payload 40%.**

## Estimate the probability of detection error (P$_E$)

The proposed multi-channel based on a deep data hiding algorithm can imperceptibly hide secret data and resist attacks even in largely hidden payloads. The algorithm's performance in data hidden can be checked by calculating the probability detection error (PE) expressed in equation Eq. (26).

$$P_E = \min\left(\frac{1}{2}(P_{FA} + P_{MD})\right). \tag{26}$$

Let the metrics (#CISI and #SICI) be the number of cover images recognized as stego-images and the number of stego-images recognized as cover images, respectively. Therefore the probability of false alarms (PFA) and miss detection (PMD) are calculated in Eqs. (27) and (28).

$$P_{FA} = \frac{\#CI_{SI}}{\#CI} \tag{27}$$

$$P_{MD} = \frac{\#SI_{CI}}{\#SI} \tag{28}$$

where $P_{FA}, P_{MD} \in [0, 1]$

The detection error (PE) value range from 0 to 0.5. When PE = 0, discovering secret data is optimal, while PE = 0.5 means perfect security and difficulty accessing secret data.

Figure 18 shows the probability detection error (PE) as a function of the payload capacity represented by the number of bits per pixel (bpp). This function measures the area under the curve (AUC), where excellent security is reached when the area of AUC is large. The proposed deep data hiding algorithm result shows that the mean value of PE equals 0.467. This result is better than the previous works (*Elshare & EL-Emam, 2018*; *Ahmad, El-Emam & AL-Azawi, 2021*) by about 3.8% and 2.2%, respectively. Moreover, the

excellent security percentage reaches 98% when bpp = 0.05, whereas the worst security percentage reaches 86% when bpp = 0.4.

## CONCLUSION AND FUTURE SCOPE

This article proposes a new steganography algorithm based on a multi-channel deep data hiding and extraction algorithm (MCDHEA). This algorithm was applied to develop the previous works (DHEA) (*Elshare & EL-Emam, 2018*) and (IDHEA) (*Ahmad, El-Emam & AL-Azawi, 2021*). The proposed hiding algorithm is based on the new approach of the modified least significant bit (NMLSB) to enhance data hiding security. However, the proposed hiding and extraction algorithm can confuse attackers because they cannot answer the questions: How many channels are used? How many levels are selected in each channel? Moreover, how is the secret message distributed on multi-channel?

The XOR encryption and Hoffman image compression were introduced to raise the security level due to the deep data hiding approach to generate one stego-image. The experimental results show that the proposed algorithm finds the best way to avoid the problem of a high load capacity of secret data. Moreover, results validate that deep dissimilarity is appreciated and obtain a more significant hiding rate. Finally, all benchmark results indicate that attackers cannot detect secret messages SM in the stego-image that appear at the last level of the main channel. The proposed deep data hiding can be enhanced in future works by applying deep learning to perform adaptive distribution of secret message SM among channels. This approach is based on partitioning SM into non-uniform sizes and randomly hiding these partitions on channels, making it hard to detect SM by attacks. This approach aims to enhance the new algorithm's performance according to the evaluation criteria for image steganography. The results show that the best PSNR and MSE obtained were 67.3 dB and 0.012, respectively, for the payload of 25,000 bytes. The best VIF and NCC obtained were 0.96 and 1, respectively, for the payload of 19,660 bytes. Finally, the best SSIM obtained was 0.999 for the payload of 294,912 bytes.

## ACKNOWLEDGEMENTS

The authors thank Dr. Gerald Schaefer and Dr. Michal Stich for publishing a very interesting UCID v2 dataset well-suited general benchmark database for evaluating and testing image hiding techniques (*Schaefer & Stich, 2004*). Moreover, the authors would like to thank Mr. Imtiaz M. S for permission to use the standard test images database for image processing.

### Funding

The publication of this research was supported by the Deanship of Scientific Research and Graduate Studies at Philadelphia University-Jordan University in Jordan. The funders had no role in study design, data collection and analysis, decision to publish, or preparation of the manuscript.

## Grant Disclosures

The following grant information was disclosed by the authors:

Deanship of Scientific Research and Graduate Studies at Philadelphia University-Jordan University in Jordan.

## Competing Interests

The authors declare that they have no competing interests.

## Author Contributions

- Hanan Hardan performed the experiments, performed the computation work, prepared figures and/or tables, and approved the final draft.
- Ali Alawneh analyzed the data, authored or reviewed drafts of the article, and approved the final draft.
- Nameer N. EL-Emam conceived and designed the experiments, authored or reviewed drafts of the article, enhancement of algorithms, and approved the final draft.

## Data Availability

The raw data is available in the Supplemental Files.

UCID data is available from Gerald Schaefer and Michal Stich "UCID: an uncompressed color image database", Proc. SPIE 5307, Storage and Retrieval Methods and Applications for Multimedia 2004, (18 December 2003); https://doi.org/10.1117/12.525375.

The standard test images are available at GitHub: https://github.com/mohammadimtiazz/standard-test-images-for-Image-Processing.git.

## Supplemental Information

Supplemental information for this article can be found online at http://dx.doi.org/10.7717/peerj-cs.1115#supplemental-information.

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
