# Peer review of "New deep data hiding and extraction algorithm using multi-channel with multi-level to improve data security and payload capacity"

_PeerJ Computer Science, doi:10.7717/peerj-cs.1115_

## Round 0.1 · original submission · Major Revisions

The paper requires major modification in the sections mentioned by the reviewers. In addition to that the paper requires proofreading and an updated reference list. You should also highlight the contribution of your research in the context of the literature.

Reviewer 1 ·

Basic reporting

1. English editing is required
2/ Literature references are insufficient, need to add some more references
3. Results should include clear definitions of all terms and theorems, and detailed proofs, which is missing

Experimental design

Research question are not well defined
Need to add some more details in the result section
Literature review part should be added to provide the current state-of-the-art

Validity of the findings

Impact and novelty not assessed properly.
Abstract and conclusion need to be refined in the light of results and contribution

Reviewer 2 ·

Basic reporting

The title needs to be rephrased possibly as, "Multi-Channels Deep Hiding and Extraction Algorithm to Improve Data Security and Payload Capacity."
Title should reflect the research carried out, and the keywords should identify the title. This would help to find this article for researchers who work in the same area. This would create more visibility and hence credibility for this research in terms of usefulness to the community and hence, citations. Please revise.

Typically, abstract is written as one single paragraph. Line # 20 and 21 of the abstract shows as two different paragraphs. Please revise.

Language in terms of reporting the research carried out is in general good, however there are some missing elements. For e.g., line # 51 and 52, "Furthermore, two methods are used to provide a higher security level and confuse attackers." If I'm not mistaken this should/could be, " Furthermore, these two methods are used to provide a higher security level and confuse attackers." Another instance is, line #201, Make the stego-image (SI) at the last level of the main channel closely match its
corresponding cover-image (CI) should be, "Make the stego-image (SI) at the last level of the main channel to closely match its corresponding cover-image (CI)." This is just a few instances that demands this manuscript needs language editing.

The abstract and introduction needs to include what are the limitations in the existing methods on information security Interms of cryptography and steganography. Hence, what is the need or motivation for this research and how this research stands out with the existing research or methods.

In Section 2 Related Work, it would be good to organize the related work in the existing literature in terms of ideas/concepts/methods etc., on their pros and cons that would justify the research reported i this manuscript. It is not a good idea to report by listing "who did what.' Related work should clearly demonstrate the shortcomings or challenges in the existing methods with a clear argument that leads to this research filling up at least some of the research gaps in the methods in the existing literature.

"Line #194 The objective of multi-channels deep hiding algorithm aims to:" This statement or phrase needs revision as objective/aim is synonymous."

Objectives stated as of line #196 to 202 is ok. However, the motivation mentioned in line # 177 to 187 must clearly explain the motivation behind this research.

Experimental design

The methodology in terms of the methods and tools used is clear, but needs language editing.

It is better to avoid personal connotations such as in line #238 In the main channel, "we" have N levels..... There are few other similar instances observed in this manuscript.

Line #290 should be, "The proposed information hiding algorithm (MCDHA) is described in the following steps:" and not 'The proposed hiding algorithm (MCDHA) is described in the following steps:'

Validity of the findings

Experimental results given in the manuscript looks credible and demonstrates the effort in terms of research carried out.

Additional comments

Experimental results given in the manuscript looks credible and demonstrates the effort in terms of research carried out. However, the presentation of the research carried out and where this research stands out when compared with the existing research needs to be clearly reported in the manuscript.

Most of the references are credible and recent, however some of the references that are more than five years could be avoided.

---

## Round 0.2 · Minor Revisions

We appreciate your response to the reviewer's comments. However, reviewers are still requesting minor revisions, so please address the comments carefully. In addition, consider the following points to highlight your research work:
1. A thorough proofread is required
2. Enhance the resolution of your pictures
3. Elaborate sufficiently about your research contributions
4. Map the research claims made in the abstract with your conclusion.

Reviewer 2 ·

Basic reporting

The Keywords following the Abstract needs revision. Should avoid too many single-words as it would make the article difficult to find during search. Should choose key word that compliments the topic.

The section "Related Works" needs rewriting.
Related work must compare, contrast, synthesize and hence, state the findings and observations made by the authors. Related Work written must 'contribute' rather than 'summarize. Writing as in line# 102 "Elshare, S., El-Emam, N. (2018) proposed a deep data hiding and extraction algorithm (DHEA)
103 to improve MLS.", then in line 108 Kordov, K., and Zhelezov, S.(2021) build an algorithm for hiding..." etc., at least to me is not a good idea as it summarizes who did what though it also highlights the pros and cons.
Related Work must connect relevant research based on common idea, theme, trend, methods, approaches etc. It would be better to organize the 'related work' in terms of methods, approaches, techniques, trend etc.
The 'related work' reported in the manuscript in essence has the essential elements but needs to be presented or reported in a fashion how related work must be. It is more of rewriting rather than redoing the related work.

Line # 184 Motivation and Objectives:
Objectives stated are clear however, the Motivation for this research is conspicuously missing.

Experimental design

No comment.

Validity of the findings

No comment.

Additional comments

Under the Acknowledgement Section, couldn't understand the purpose of the statement "The authors thank Dr. Gerald Schaefer and Dr. Michal Stich for publishing a very interested 747 UCID v2 dataset well-suited general benchmark database for evaluating and testing image hiding 748 techniques (Schaefer, G., Stich, M. 2004). " and it's relevance to this manuscript.

---

## Round 0.3 · accepted · Accept

Thanks for addressing the comments critically, the manuscript now stands for acceptance.